# Principle of Relevant Information for Graph Sparsification

**Shujian Yu**[1]  **Francesco Alesiani**[2]  **Wenzhe Yin**[3]  **Robert Jenssen**[1,5,6]  **Jose C. Principe**[4]

[1]UiT - The Arctic University of Norway, Norway
[2]NEC Laboratories Europe, Germany
[3]University of Amsterdam, Netherlands
[4]University of Florida, USA
[5]Norwegian Computing Center, Norway
[6]University of Copenhagen, Denmark

## Abstract

Graph sparsification aims to reduce the number of edges of a graph while maintaining its structural properties. In this paper, we propose the first general and effective information-theoretic formulation of graph sparsification, by taking inspiration from the Principle of Relevant Information (PRI). To this end, we extend the PRI from a standard scalar random variable setting to structured data (i.e., graphs). Our Graph-PRI objective is achieved by operating on the graph Laplacian, made possible by expressing the graph Laplacian of a subgraph in terms of a sparse edge selection vector **w**. We provide both theoretical and empirical justifications on the validity of our Graph-PRI approach. We also analyze its analytical solutions in a few special cases. We finally present three representative real-world applications, namely graph sparsification, graph regularized multi-task learning, and medical imaging-derived brain network classification, to demonstrate the effectiveness, the versatility and the enhanced interpretability of our approach over prevalent sparsification techniques. Code of Graph-PRI is available at https://github.com/SJYuCNEL/PRI-Graphs.

## 1 INTRODUCTION

Many complex structures and phenomena are naturally described as graphs and networks (e.g., social networks, brain functional connectivity [Zhou et al., 2020], climate causal effect network [Nowack et al., 2020], etc.). However, it is challenging to exactly visualize and analyze a graph even with moderate size due to the quadratic growth in the number of edges. Therefore, techniques to sparsify graphs by pruning less informative edges have gained increasing attention in the last two decades [Spielman and Srivastava,

2011, Bravo Hermsdorff and Gunderson, 2019, Wu and Chen, 2020]. Apart from offering a much easier visualization, graph sparsification can be used in multiple ways. For example, it may reduce the storage space and accelerate the running time of machine learning algorithms involving graph regularization, with negligible accuracy loss [Sadhanala et al., 2016]. When differentiable privacy is a major concern, sparsity can remove or hide edges for the purpose of information protection [Arora and Upadhyay, 2019].

On the other hand, there is a recent trend to leverage information-theoretic concepts and principles to problems related to graphs or graph neural networks. Let $\mathcal{X}$ denote the graph input data which may encode both graph structure information (characterized by either adjacency matrix $A$ or graph Laplacian $L$) and node attributes, and $Y$ the desired response such as node labels or graph labels. A notable example is the famed Information Bottleneck (IB) approach [Tishby et al., 1999], which formulates the learning as:

$$\mathcal{L}_{\text{IB}} = \min I(\mathcal{X}; T) - \beta I(Y; T), \qquad (1)$$

in which $I(\cdot; \cdot)$ denotes the mutual information. $T$ is the object we want to learn or infer from $\{\mathcal{X}, Y\}$ that can be used as graph node representation [Wu et al., 2020] or as the most infromative and interpretable subgraph with respect to the label $Y$ [Yu et al., 2020]. $\beta$ is a Lagrange multiplier that controls the trade-off between the **sufficiency** (the performance of $T$ on down-stream task, as quantified by $I(Y; T)$) and the **minimality** (the complexity of the representation, as measured by $I(\mathcal{X}; T)$).

Instead of using the IB approach, we explore the feasibility and power in graph data of another less well-known information-theoretic principle - the Principle of Relevant Information (PRI) [Principe, 2010, Chapter 8], which exploits self organization, requiring only a single random variable $\mathcal{X}$. Different from IB that requires an auxiliary relevant variable $Y$ and possibly the joint distribution of $\mathbb{P}(\mathcal{X}, Y)$, the PRI is fully unsupervised and aims to obtain a reduced

*Accepted for the 38*th *Conference on Uncertainty in Artificial Intelligence* (UAI 2022).

statistical representation $T$ by decomposing $\mathcal{X}$ with:

$$\mathcal{L}_{\text{PRI}} = \min H(T) + \beta D(\mathbb{P}(\mathcal{X})\|\mathbb{P}(T)), \qquad (2)$$

where $H(T)$ refers to the entropy of $T$. The minimization of entropy can be viewed as a means of reducing uncertainty and finding the statistical **regularity** in $T$. $D(\mathbb{P}(\mathcal{X})\|\mathbb{P}(T))$ is the divergence between the distributions of $\mathcal{X}$ (i.e., $\mathbb{P}(\mathcal{X})$) and $T$ (i.e., $\mathbb{P}(T)$), which quantifies the **descriptive power** of $T$ about $\mathcal{X}$.

So far, PRI has only been used in a standard scalar random variable setting. Recent applications of PRI include, but are not limited to, selecting the most relevant examples from the majority class in imbalanced classification [Hoyos-Osorio et al., 2021], and learning disentengled representations with variational autoencoder [Li et al., 2020]. Usually, one uses the 2-order Rényi's entropy [Rényi, 1961] to quantify $H(T)$ and the Cauchy-Schwarz (CS) divergence [Jenssen et al., 2006] to quantify $D(\mathbb{P}(X)\|\mathbb{P}(T))$ for ease of optimization.

In this paper, we extend PRI to graph data. This is not a trivial task, as the Rényi's quadratic entropy and CS divergence are defined over probability space and do not capture any structure information. We also exemplify our Graph-PRI with an application in graph sparsification. To summarize, our contributions are fourfold:

- Taking the graph Laplacian as the input variable, we propose a new information-theoretic formulation for graph sparsification, by taking inspiration from PRI.

- We provide theoretical and empirical justifications to the objective of Graph-PRI for sparsification. We also analyze the analytical solutions in some special cases of hyperparameter $\beta$.

- We demonstrate that the graph Laplacian of the resulting subgraph can be elegantly expressed in terms of a sparse edge selection vector $\mathbf{w}$, which significantly simplify the learning argument of Graph-PRI. We also show that the objective of Graph-PRI is differentiable, which further simplifies the optimization.

- Experimental results on graph sparsification, graph-regularized multi-task learning, and brain network classification demonstrate the versatility and compelling performance of Graph-PRI.

## 2 PRELIMINARY KNOWLEDGE

### 2.1 PROBLEM DEFINITION AND NOTATIONS

Consider an undirected graph $G = (V, E)$ with a set of nodes $V = \{v_1, \cdots, v_N\}$ and a set of edges $E = \{e_1, \cdots, e_M\}$ which reveals the connections between nodes. The objective of graph sparsification is to preferentially retain a small subset of edges from $G$ to obtain a sparsified surrogate graph $G_s = (V, E_s)$ with the edge set $E_s \subset E$ such

that $|E_s| \ll M$ [Spielman and Srivastava, 2011, Hamann et al., 2016].

Alternatively to graph sparsification, it is also possible to reduce the nodes of the graph, which is called graph coarsening. Recent examples on graph coarsening include [Loukas, 2019, Cai et al., 2021]. Recently, [Bravo Hermsdorff and Gunderson, 2019] provides a unified framework for graph sparsification and graph coarsening. In this work, we only focus on graph sparsification.

The topology of $G$ is essentially determined by its graph Laplacian $L = D - A$, where $A$ is the adjacency matrix and $D = \text{diag}(\mathbf{d})$ is the diagonal matrix formed by the degrees of the vertices $d_i = \sum_{j=1}^{N} A_{ij}$. Consider an arbitrary orientation of edges of $G$, the incidence matrix $B = [\mathbf{b}_1, \cdots, \mathbf{b}_M]$ of $G$ is a $N \times M$ matrix whose entries is given by:

$$[\mathbf{b}_m]_i = \begin{cases} +1 & \text{if node } v_i \text{ is the head of edge } e_m \\ -1 & \text{if node } v_i \text{ is the tail of edge } e_m \\ 0 & \text{otherwise} \end{cases} \quad (3)$$

Mathematically, $L$ can be expressed in terms of $B$ as:

$$L = BB^T = \sum_{m=1}^{M} \mathbf{b}_m \mathbf{b}_m^T. \qquad (4)$$

Suppose the subgraph $G_s$ contains $K$ edges, one can obtain $G_s$ from $G$ through an edge selection vector $\mathbf{w} = [w_1, \cdots, w_M]^T \in \{0, 1\}^M$. Here, $\|\mathbf{w}\|_0 = K$, $w_m = 1$ if the $m$-th edge belongs to the edge subset $E_s$, and $w_m = 0$ otherwise. Finally, one can write the graph Laplacian $L_s$ of the subgraph $G_s$ as a function of $\mathbf{w}$ by the following formula:

$$L_s(\mathbf{w}) = \sum_{m=1}^{M} w_m \mathbf{b}_m \mathbf{b}_m^T = B \, \text{diag}(\mathbf{w}) B^T, \qquad (5)$$

in which $\text{diag}(\mathbf{w}) \in \mathbb{R}^{M \times M}$ is a square diagonal matrix with $\mathbf{w}$ on the main diagonal.

Note that, Eq. (5) also applies to weighted graph $G = (V, W)$ by a proper reformulation of the incidence matrix $B$ as:

$$[\mathbf{b}_m]_i = \begin{cases} +\sqrt{\mu_m} & \text{if node } v_i \text{ is the head of edge } e_m \\ -\sqrt{\mu_m} & \text{if node } v_i \text{ is the tail of edge } e_m \\ 0 & \text{otherwise} \end{cases} ,$$

$$(6)$$

in which $\mu_m$ is the weight of edge $e_m$.

In what follows, we will design a learning-based approach to optimally obtain the edge selection vector $\mathbf{w}$ by making use of the general idea of PRI.

## 2.2 GRAPH SPARSIFICATION

Substantial efforts have been made on graph sparsification. In general, existing methods are mostly based on sampling Fung et al. [2019], Wickman et al. [2021], in which the importance of edges can be evaluated by effective resistance [Spielman and Srivastava, 2011, Spielman and Teng, 2011], degree of neighboring nodes [Hamann et al., 2016] or local similarity [Satuluri et al., 2011]. Among them, the most notable example is the spectrum-preserving approach that generates a $\gamma$-spectral approximation to $G$ such that:

$$\frac{1}{\gamma}\vec{x}^T L_s \vec{x} \leq \vec{x}^T L \vec{x} \leq \gamma \vec{x}^T L_s \vec{x} \quad \text{for all} \quad \vec{x}. \quad (7)$$

Remarkably, Spielman *et al.* also proved that every graph $G$ has an $(1+\epsilon)$-spectral approximation $G_s$ with nearly $\mathcal{O}(\frac{N}{\epsilon^2})$ edges.

Learning-based approach (especially which uses neural networks) for graph sparsification, in which there is an explicit learning objective and can be directed optimized, is still less-investigated. GSGAN [Wu and Chen, 2020] is designed mainly for community detection, whereas SparRL [Wickman et al., 2021] uses deep reinforcement learning to sequentially prune edges by preserving the subgraph modularity. Different from GSGAN and SparRL, we demonstrate below that a sparsified graph can be learned simply by a gradient-based method in a principled (information-theoretic) manner, avoiding the necessity of reinforcement learning or the tuning of a generative adversarial network (GAN) [Goodfellow et al., 2014].

# 3 PRI FOR GRAPH SPARSIFICATION

## 3.1 THE LEARNING OBJECTIVE

Suppose we are given a graph $G$ with a known but fixed topology that is characterized by its graph Laplacian $\rho$, from which we want to obtain a surrogate subgraph $G_s$ with graph Laplacian $\sigma$, by preferentially removing less informative (or redundant) edges in $G$. Motivated by the objective of PRI in Eq. (2), we can cast this problem as a trade-off between the entropy $S(\sigma)$ of $G_s$ and its descriptive power about $G$ in terms of their divergence (or dissimilarity) $D(\sigma||\rho)$:

$$\mathcal{J}_{\text{Graph-PRI}} = \arg\min_{\sigma} S(\sigma) + \beta D(\sigma||\rho), \quad (8)$$

In this paper, we choose von Neumann entropy on the trace normalized graph Laplacian (i.e., $\tilde{\sigma} = \sigma/\operatorname{tr}(\sigma)$ to quantify the entropy of $G_s$, which is defined on the cone of symmetric positive semi-definite (SPS) matrix with trace 1 as [Nielsen and Chuang, 2002]:

$$S_{\text{vN}}(\tilde{\sigma}) = -\operatorname{tr}(\tilde{\sigma}\log\tilde{\sigma}) = -\sum_i (\lambda_i \log \lambda_i), \quad (9)$$

in which $\log(\cdot)$ is the matrix logarithm, $\operatorname{tr}(\cdot)$ denotes the trace, $\{\lambda_i\}$ are the eigenvalues of $\tilde{\sigma}$.

We then use the quantum Jenssen-Shannon (QJS) divergence between two trace normalized graph Laplacians $\tilde{\sigma}$ and $\tilde{\rho}$ to quantify the divergence between $G$ and $G_s$ [Lamberti et al., 2008]:

$$D_{\text{QJS}}(\tilde{\sigma}||\tilde{\rho}) = S_{\text{vN}}\left(\frac{\tilde{\sigma}+\tilde{\rho}}{2}\right) - \frac{1}{2}S_{\text{vN}}(\tilde{\sigma}) - \frac{1}{2}S_{\text{vN}}(\tilde{\rho}). \quad (10)$$

In this paper, we absorb a scaling constant 2 into the expression for $D_{\text{QJS}}(\tilde{\sigma}||\tilde{\rho})$, the resulting objective combining Eqs. (8)-(10) is given by:

$$\begin{aligned}
\mathcal{J}_{\text{Graph-PRI}} &= \arg\min S_{\text{vN}}(\tilde{\sigma}) + \beta D_{\text{QJS}}(\tilde{\sigma}||\tilde{\rho}) \quad (11)\\
&= \arg\min S_{\text{vN}}(\tilde{\sigma})\\
&\quad + \beta\left[2S_{\text{vN}}\left(\frac{\tilde{\sigma}+\tilde{\rho}}{2}\right) - S_{\text{vN}}(\tilde{\sigma}) - S_{\text{vN}}(\tilde{\rho})\right]\\
&\equiv \arg\min(1-\beta)S_{\text{vN}}(\tilde{\sigma}) + 2\beta S_{\text{vN}}\left(\frac{\tilde{\sigma}+\tilde{\rho}}{2}\right).
\end{aligned}$$

We remove an extra term $-\beta S_{\text{vN}}(\tilde{\rho})$ in the last line of Eq. (11), because it is a constant value with respect to $\tilde{\sigma}$.

## 3.2 JUSTIFICATION OF THE OBJECTIVE OF GRAPH-PRI

One may ask why we choose the von Neumann entropy in $\mathcal{J}_{\text{graph-PRI}}$. In fact, the Laplacian spectrum contains rich information about the multi-scale structure of graphs [Mohar, 1997]. For example, it is well-known that the second smallest eigenvalue $\lambda_2(L)$, which is also called the algebraic connectivity, is always considered to be a measure of how well-connected a graph is [Ghosh and Boyd, 2006].

On the other hand, it is natural to use the QJS divergence to quantify the dissimilarity between the original graph and its sparsified version. The QJS divergence is symmetric and its square root has also recently been found to satisfy the triangle inequality [Virosztek, 2021]. In fact, as a graph dissimilarity measure, QJS has also found applications in multilayer networks compression [De Domenico et al., 2015] and anomaly detection in graph streams [Chen et al., 2019].

A few recent studies indicate the close connections between $S_{\text{vN}}(L)$ with the structure *regularity* and *sparsity* of a graph [Passerini and Severini, 2008, Han et al., 2012, Liu et al., 2021, Simmons et al., 2018]. We shall now highlight three theorems therein and explain our justifications in Sections 3.2.1 and 3.2.2 in detail.

**Theorem 1** ([Passerini and Severini, 2008]). *Given an undirected graph* $G = \{V, E\}$, *let* $G' = G + \{u, v\}$, *where* $V(G) = V(G')$ *and* $E(G) = E(G') \cup \{u, v\}$, *we have:*

$$S_{\text{vN}}(L_{G'}) \geq \frac{d_{G'} - 2}{d_{G'}} S_{\text{vN}}(L_G), \quad (12)$$

where $d_{G'} = \sum_{v \in V(G')} d(v)$ is the degree-sum of $G'$, $L_G$ and $L_{G'}$ refer to respectively the graph Laplacians of $G$ and $G'$.

Theorem 1 shows that $S_{vN}(L)$ tends to grow with edge addition. Although Eq. (12) does not indicate a monotonic increasing trend for $S_{vN}(L)$, it does suggest that minimizing $S_{vN}(L)$ may lead to a sparser graph, especially when the degree-sum is large.

**Theorem 2** ([Liu et al., 2021]). *For any undirected graph $G = \{V, E\}$, we have:*

$$0 \leq \Delta H(G) = H(G) - S_{vN}(L_G) \leq \frac{\log_2 e}{\delta} \frac{\mathrm{tr}(W^2)}{d_G},$$

(13)

*where $H(G) = -\sum_{i=1}^{N} \left(\frac{d_i}{d_G}\right) \log_2 \left(\frac{d_i}{d_G}\right)$, in which $\delta = \min d_i | d_i > 0$ is the minimum positive node degree, $d_G$ is the degree-sum, $W$ is the weighted adjacency matrix of $G$.*

**Theorem 3** ([Liu et al., 2021]). *For almost all unweighted graphs $G$ of order $n$, we have:*

$$\frac{H(G)}{S_{vN}(L_G)} - 1 \geq 0,$$

(14)

*and decays to 0 at a rate of $\mathcal{O}(1/\log_2(n))$.*

Theorem 2 and Theorem 3 bound the difference between $S_{vN}(L)$ and $H(G)$, the Shannon discrete entropy on node degree. They also indicate that $H(G)$ is a natural choice of the fast approximation to $S_{vN}(L)$. In fact, there are different fast approximation approaches so far [Chen et al., 2019, Minello et al., 2019, Kontopoulou et al., 2020]. According to [Liu et al., 2021], $H(G)$ enjoys simultaneously good scalability, interpretability and provable accuracy.

### 3.2.1 $\beta$ controls the sparsity of $G_s$

Different from the spectral sparsifiers [Spielman and Srivastava, 2011, Spielman and Teng, 2011] in which the sparsity of the subgraph is hard to control (i.e., there is no monotonic relationship between the hyperparameter $\epsilon$ and the degree of sparsity as measured by $|E_s|$), we argue that the sparsity of subgraph obtained by Graph-PRI is mainly determined by the value of hyperparameter $\beta$.

Our argument is mainly based on Theorem 1. Here, we additionally claim that, under a mild condition (Assumption 1), the QJS divergence $D_{\mathrm{QJS}}(L \| L_s)$ is prone to decrease with edge addition (Corollary 1).

**Assumption 1.** *Given an undirected graph $G = \{V, E\}$, let $G' = G + \{u, v\}$, where $V(G) = V(G')$ and $E(G) = E(G') \cup \{u, v\}$, we have $S_{vN}(L_{G'}) \geq S_{vN}(L_G)$, i.e., there exists a strictly monotonically increasing relationship between the number of edges $|G|$ and the von Neumann entropy $S_{vN}(L_G)$.*

**Corollary 1.** *Under Assumption 1, suppose $G_s = \{V_s, E_s\}$ is a sparse graph obtained from $G = \{V, E\}$ (by removing edges), let $G'_s = G_s + \{u, v\}$, where $\{u, v\}$ is an edge from the original graph $G$, $V(G_s) = V(G'_s)$ and $E(G_s) = E(G'_s) \cup \{u, v\}$, we have $D_{QJS}(L_{G'_s} \| L_G) \leq D_{QJS}(L_{G_s} \| L_G)$, i.e., adding an edge is prone to decrease the QJS divergence.*

We provide additional information on the rigor of Assumption 1 in Appendix A.1. Combining Theorem 1 and Corollary 1, it is interesting to find that the edge addition has opposite effects on $S_{vN}$ and $D_{QJS}$: the former is likely to increase whereas the latter will decrease. Therefore, when minimizing the weighted sum of $S_{vN}$ and $D_{QJS}$ together as in Graph-PRI, one can expect the number of edges in $G_s$ is mainly determined by the hyperprameter $\beta$: a smaller $\beta$ gives more weight to $S_{vN}$ and thus encourages a more sparse graph.

To empirically justify our argument, we generate a set of graphs with 200 nodes by either the Erdös-Rényi (ER) model or the Barabási-Albert (BA) model [Barabási and Albert, 1999]. For both models, we generate the original dense graph $G$ where the average of the node degree $\bar{d}$ is approximately 10, 20 and 30, respectively. We then sparsify $G$ to obtain $G_s$ by a random sparsifier, which satisfies the spectral property (i.e., Eq. (7)), whose computational complexity is, however, low [Sadhanala et al., 2016].

We finally evaluate the von Neumann entropy of $G_s$ and the QJS divergence between $G$ and $G_s$ with respect to different percentages (pct.) of preserved edges. We repeat the procedure 100 independent times and the averaged results are plotted in Fig. 1, from which we can clearly observe the opposite effects mentioned above. We also sparisify the original graph $G$ by our Graph-PRI with different values of $\beta$. The number of preserved edges in $G_s$ with respect to $\beta$ is illustrated in Fig. 2, from which we can observe an obvious monotonic increasing relationship.

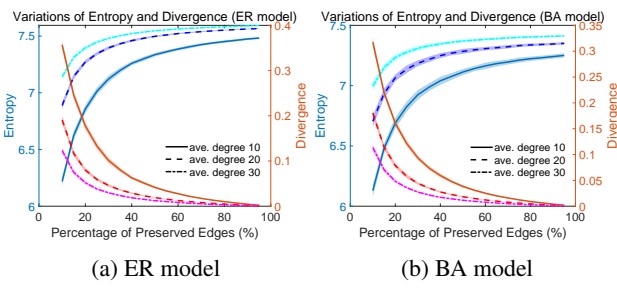

(a) ER model        (b) BA model

Figure 1: The variations of entropy and divergence with respect to different percentages of preserved edges.

### 3.2.2 Graph-PRI in special cases of $\beta$

Continuing our discussion in Sec. 3.2.1, it would be interesting to infer what may happen in some special cases of $\beta$.

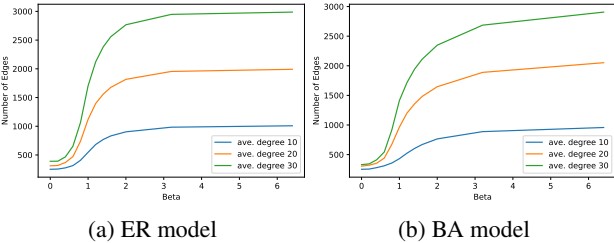

(a) ER model  (b) BA model

Figure 2: The monotonic increasing relationship between number of preserved edges and hyperparameter $\beta$ in our Graph-PRI.

Here, we restrict our discussion with $\beta = 0$ and $\beta \to \infty$.

When $\beta = 0$, due to Theorems 2 and 3, our objective can be interpreted as $\min H(G)$. $H(G)$ takes the mathematical form of the Shannon discrete entropy (i.e., $-\sum_{i=1}^{N} \mathbb{P}(x_i) \log \mathbb{P}(x_i)$, in which $\mathbb{P}(x_i)$ is the probability of the $i$-th state) on the degree of node. In this sense, $H(G)$ reaches to maximum for uniformly distributed degree of node (i.e., $d_1 = \cdots = d_N = k$, which is also called the $k$-regular graph) and reduces to minimum if the degree of one node dominates (i.e., a star graph that possesses a high level of centralization). In fact, it was also conjectured in [Dairyko et al., 2017] that among connected graphs with fixed order $n$, the star graph $S_n$ minimizes the von Neumann entropy. Thus, $S_{vN}(L)$ can also be interpreted as a measure of degree heterogeneity or graph centrality [Simmons et al., 2018]. It also indicates that minimizing $S_{vN}(L)$ pushes the Graph-PRI to learn a graph that has more graph centrality.

When $\beta \to \infty$, we are expect to recover original graph $G$ by Corollary 1. Fig. 3 corroborates our analysis.

Interestingly, similar properties also hold for the original PRI in scalar random variable setting (see Appendix B).

### 3.3  OPTIMIZATION

We define a gradient descendent algorithm to solve Eq. (11). As has been discussed in Section 2.1, we have $\rho = BB^T$ and $\sigma_\mathbf{w} = B \operatorname{diag}(\mathbf{w}) B^T$, in which $\mathbf{w}$ is the edge selection vector. For simplicity, we assume that the selections of edges from the original graph $G$ are conditionally independent to each other [Luo et al., 2020], that is $\mathbb{P}_\mathbf{w} = \prod_{i=1}^{M} \mathbb{P}_{w_i}$. Due to the discrete nature of $G_s$, we relax $\mathbf{w} = [w_1, w_2, ..., w_M]$ from a binary vector $\{0, 1\}^M$ to a continuous real-valued vector in $[0, 1]^M$. In this sense, the value of $w_i$ can be interpreted as the probability of selecting the $i$-th edge.

In practice, we use the Gumbel-softmax [Maddison et al., 2017, Jang et al., 2016] to update $w_i$. Particularly, suppose we want to approximate a categorical random variable represented as a one-hot vector in $\mathbb{R}^K$ with category probability $p_1, p_2, \cdots, p_K$ (here, $K = 2$), the Gumbel-softmax gives a

$K$-dimensional sampled vector with the $i$-th entry as:

$$\hat{p}_i = \frac{\exp\left((\log p_i + g_i)/\tau\right)}{\sum_{j=1}^{K} \exp\left((\log p_j + g_j)/\tau\right)}, \qquad (15)$$

where $\tau$ is a temperature for the Concrete distribution and $g_i$ is generated from a Gumbel$(0, 1)$ distribution:

$$g_i = -\log(-\log u_i), \quad u_i \sim \text{Uniform}(0, 1). \qquad (16)$$

Note that, although we use the Gumbel-Softmax to ease the optimization, Graph-PRI itself has analytical gradient (Theorem 4). The detailed algorithm of Graph-PRI is elaborated in Appendix E. We also provide a PyTorch example therein.

**Theorem 4.** *The gradient of Eq. (11) with respect to edge selection vector $\mathbf{w}$ is:*

$$\nabla_\mathbf{w} \mathcal{J}_{\textit{Graph-PRI}} = U g, \qquad (17)$$

*where $\tilde{\mathbf{w}}$ is the normalised $\mathbf{w}$ ($\tilde{\mathbf{w}} = \mathbf{w}/\sum_{i=1}^{M} w_i$), $\tilde{\mathbf{1}}_M = \frac{1}{M} \mathbf{1}_M$ is the normalized version of the all-ones vector. $\bar{\sigma}_\mathbf{w} = \frac{1}{2}(\tilde{\sigma}_\mathbf{w} + \tilde{\rho}) = \frac{1}{2} B \operatorname{diag}(\tilde{\mathbf{w}} + \tilde{\mathbf{1}}_M) B^T$. $g = -\operatorname{diag}\left(B^T[(1-\beta)\ln\tilde{\sigma}_\mathbf{w} + \beta\ln\bar{\sigma}_\mathbf{w}]B\right)$ and $U = \{u_{ij}\} \in \mathbb{R}^{M \times M}, u_{ij} = -\frac{\tilde{w}_j}{1-\tilde{w}_i}, \forall ij|i \neq j, u_{ii} = 1$.*

### 3.4  APPROXIMATION AND CONNECTIVITY CONSTRAINT

The computation of von Neumann entropy requires the eigenvalue decomposition of a trace normalized SPS matrix, which usually takes $\mathcal{O}(N^3)$ time. In practical applications in which the computational time is a major concern (i.e., when training deep neural networks or when dealing with large graphs with hundreds of thousands of nodes), based on Theorems 2 and 3, we simply approximate $S_{vN}(L_G)$ with the Shannon discrete entropy on the normalized degree of nodes $H(G)$, which immediately reduces the computational complexity to $\mathcal{O}(N)$. Unless otherwise specified, the experiments in the next section still use the basic $S_{vN}(L_G)$.

On the other hand, when the connectivity of the subgraph is preferred, one can simply add another regularization on the degree of the nodes [Kalofolias, 2016]:

$$\min_\mathbf{w} S(\tilde{\sigma}_\mathbf{w}) + \beta D(\tilde{\sigma}_\mathbf{w}||\tilde{\rho}) - \alpha \mathbf{1}^T \log(\operatorname{diag}(\sigma)), \qquad (18)$$

where the hyper-parameter $\alpha > 0$. This Logarithm barrier forces the degree to be positive and improves the connectivety of graph without compromising sparsity. Unless otherwise specified, we select $\alpha = 0.005$ throughout this work.

## 4  EXPERIMENTAL EVALUATION

In this section, we demonstrate the effectiveness and versatility of our Graph-PRI in multiple graph-related machine

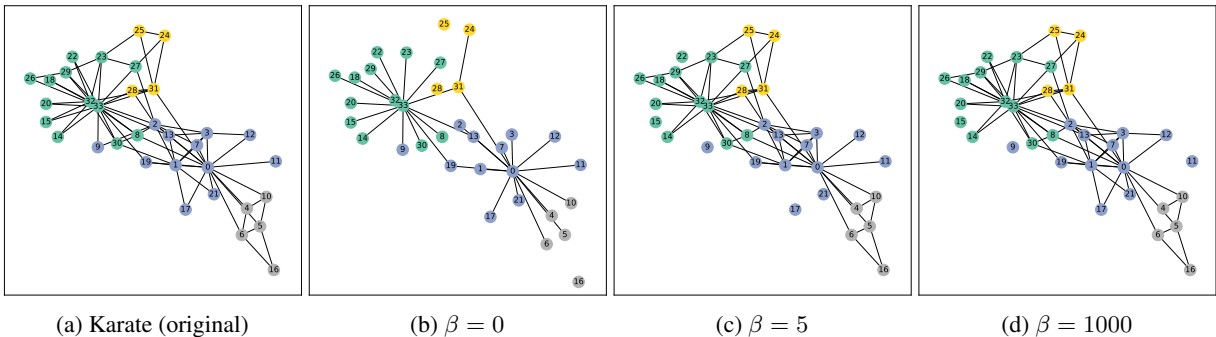

| (a) Karate (original) | (b) $\beta = 0$ | (c) $\beta = 5$ | (d) $\beta = 1000$ |

Figure 3: Illustration of the sparsified graph structures revealed by our Graph-PRI for (a) Zachary's karate club Zachary [1977]. As the values of $\beta$ increases, the solution passes through (b) an approximately star graph to the extreme case of (d) $\beta \to \infty$, in which we get back the original graph as the solution.

learning tasks. Our experimental study is guided by the following three questions:

**Q1** What kind of structural property or information does our method preserves?

**Q2** How well does our method compare against popular and competitive graph sparsification baselines?

**Q3** How to use the Graph-PRI in practical machine learning problems; and what are the performance gains?

The selected competing methods include 1 baseline and 3 state-of-the-art (SOTA) ones: 1) the Random Sampling (RS) that randomly prunes a percentage of edges; 2) the Local Degree (LD) [Hamann et al., 2016] that only preserves the top $|\text{degree}(v)^\alpha|$ ($0 \le \alpha \le 1$) neighbors (sorted by degree in descending order) for each node $v$; 3) the Local Similarity (LS) [Satuluri et al., 2011]) that applies Jaccard similarity function on nodes $v$ and $u$'s neighborhoods to quantify the score of edge $(u, v)$; 4) the Effective Resistance (ER) [Spielman and Srivastava, 2011]. We implement RS, LD, LS by NetworKit[1], and ER by PyGSP[2].

## 4.1 GRAPH SPARSIFICATION

We use 2 synthetic data and 4 real-world network data from KONECT network datasets[3] for evaluation. They are, **G1**: a $k$-NN ($k = 10$) graph with 20 nodes that constitute a global circle structure; **G2**: a stochastic block model (SBM) with four distinct communities (30 nodes in each community, and intra- and inter-community connection probabilities of $2^{-2}$ and $2^{-7}$, respectively); **G3**: the most widely used Zachary karate club network (34 nodes and 78 edges); **G4**: a network contains contacts between suspected terrorists involved in the train bombing of Madrid on March 11, 2004 (64 nodes and 243 edges); **G5**: a network of books about US politics published around the time of the 2004 presidential election

---

[1]https://networkit.github.io/
[2]https://github.com/epfl-lts2/pygsp
[3]http://konect.cc/networks/

and sold by the online bookseller Amazon.com (105 nodes and 441 edges); and **G6**: a collaboration network of Jazz musicians (198 nodes and 2, 742 edges).

We expect Graph-PRI to preserve two essential properties associated with the original graph: 1) the spectral similarity (due to the divergence term); and 2) the graph centrality (due to the entropy term). We empirically justify our claims with two metrics. They are, the geodesic distance $d_{\vec{x}}(\rho, \sigma)$ [Bravo Hermsdorff and Gunderson, 2019]:

$$d_{\vec{x}}(\rho, \sigma) = \text{arccosh}\left(1 + \frac{\|(\rho - \sigma)\vec{x}\|_2^2 \|\vec{x}\|_2^2}{2(\vec{x}^T \rho \vec{x})(\vec{x}^T \sigma \vec{x})}\right), \quad (19)$$

in which we select $\vec{x}$ to be the smallest non-trivial eigenvector of the original Laplacian $\rho$, as it encodes the global structure of a graph; and the graph centralization measure by $C_D$ [Freeman, 1978]):

$$C_D = \frac{\sum_{i=1}^N \max(d_j) - d_i}{N^2 - 3N + 2}, \quad (20)$$

in which $\max(d_j)$ refers to the maximum node degree.

We demonstrate in Fig. 4 and Fig. 5 respectively the values of $d_{\vec{x}}(\rho, \sigma)$ and $C_D$ with respect to different edge preserving ratio (i.e., $|E_s|/|E|$) for different sparsification methods. As can be seen, our Graph-PRI always achieves the 2nd best performance across different graphs. Although LD has advantages on preserving spectral distance and graph centrality, it does not have compelling performance in practical applications as will be demonstrated in the next subsection.

## 4.2 GRAPH-REGULARIZED MULTI-TASK LEARNING

In traditional multi-task learning (MTL), we are given a group of $T$ related tasks. In each task we have access to a training set $\mathcal{D}_t$ with $N_t$ data instances $\{(\mathbf{x}_t^i, y_t^i) : i = 1, \cdots, N_t, t = 1, \cdots, T\}$. In this section, we focus on the regression setup in which $\mathbf{x}_t^i \in \mathcal{X}_t \subseteq \mathbb{R}^d$ and $y_t^i \in \mathbb{R}$. Multi-task learning aims to learn from each training set $\mathcal{D}_t$

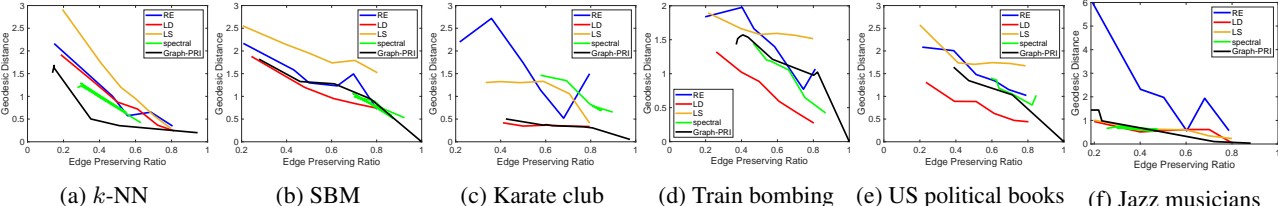

| (a) $k$-NN | (b) SBM | (c) Karate club | (d) Train bombing | (e) US political books | (f) Jazz musicians |

Figure 4: Spectral distance $d_{\vec{x}}(\rho, \sigma)$ (the smaller the better).

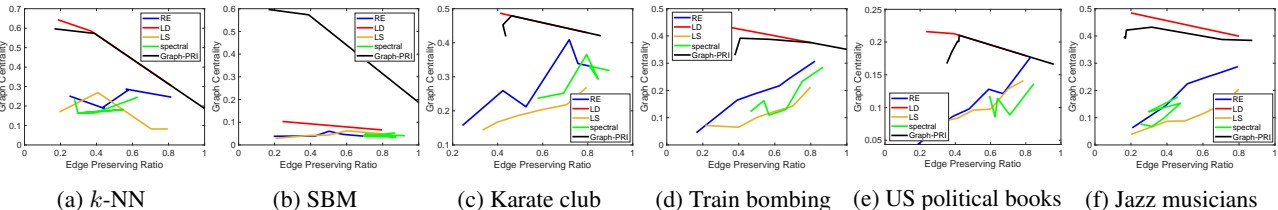

| (a) $k$-NN | (b) SBM | (c) Karate club | (d) Train bombing | (e) US political books | (f) Jazz musicians |

Figure 5: Graph centrality $C_D$ (the larger the higher degree of centrality).

a prediction model $f_t(\mathbf{w}_t, \cdot) : \mathcal{X}_t \to \mathbb{R}$ with parameter $\mathbf{w}_t$ such that the task relatedness is taken into consideration and the overall generalization error is small.

In what follows, we assume a linear model in each task, i.e., $f_t(\mathbf{w}_t, \mathbf{x}) = \mathbf{w}_t^T \mathbf{x}$. The multi-task regression problem with a regularization $\Omega$ on the model parameters $W = [\mathbf{w}_1, \mathbf{w}_2, \cdots, \mathbf{w}_T]$ can thus be defined as:

$$\min_W \sum_{t=1}^{T} \|\mathbf{w}_t^T \mathbf{x}_t - y_t\|_2^2 + \gamma \Omega(W). \quad (21)$$

Graph is a natural way to establish the relationship over multiple tasks: each node refers to a single task; if two tasks are strongly correlated to each other, there is an edge to connect them. In this sense, the objective for multi-task regression learning regularized with a graph adjacency matrix $A$ can be formulated as [He et al., 2019]:

$$\min_W \sum_{t=1}^{T} \|\mathbf{w}_t^T \mathbf{x}_t - y_t\|_2^2 + \gamma \sum_{i=1}^{T} \sum_{j \in \mathcal{N}_i} A_{ij} \|\mathbf{w}_i - \mathbf{w}_j\|_2^2, \quad (22)$$

where $\mathcal{N}_i$ is the set of neighbors of $i$-th task.

Usually, a dense graph $G$ is estimated at first to fully characterize task relatedness [Chen et al., 2010, He et al., 2019]. Here, we are interesting in: 1) sparsifying $G$ to reduce redundant or less-important connections (edges) between tasks; and 2) validating if the sparsified graph can further reduce the generalization error.

To this end, we exemplify our motivation with the recently proposed Convex Clustering Multi-Task regression Learning (CCMTL) [He et al., 2019] that optimizes Eq. (22) with the Combinatorial Multigrid (CMG) solver [Koutis et al., 2011], and test its performance on two benchmark MTL datasets[4]: 1) a synthetic dataset Gonçalves et al. [2016]

---

[4]See Appendix C on details of datasets in sections 4.2 and 4.3.

with 20 tasks in which tasks 1-10 are mutually related and tasks 11-20 are mutually related; 2) a real-world Parkinson's disease dataset[5] which contains biomedical voice measurements from 42 patients. We view each patient as a single task and aim to predict the motor Unified Parkinson's Disease Rating Scale (UPDRS) score based 19-dimensional features such as age, gender, and jitter and shimmer voice measurements. In both datasets, the initial dense task-relatedness graph $G$ is estimated in the following way: we perform linear regression on each task individually; the task-relatedness between two tasks is modeled as the $\ell_2$ distance of their independently learned linear regression coefficients; we then construct a $k$-nearest neighbor ($k = 10$) graph based on all pairwise task distances as $G$.

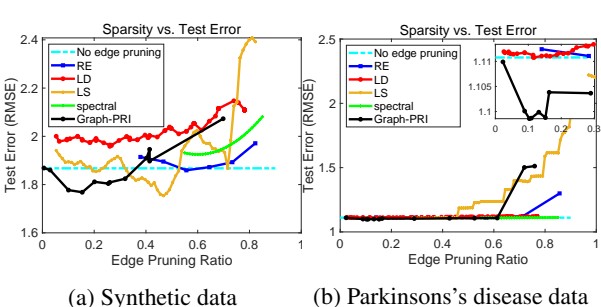

| (a) Synthetic data | (b) Parkinsons's disease data |

Figure 6: The RMSE with respect to the degree of sparsity (defined as $1 - |E_s|/|E|$) of the resulting subgraph for all competing methods. Black dashed line indicates performance without any edge pruning. Our method is able to drop out redundant or less-important edges to further reduce generalization error.

We evaluate the test performance with the root mean squared error (RMSE) and demonstrate the values of RMSE with respect to different edge pruning ratio (i.e., $1 - |E_s|/|E|$) of different methods in Fig. 6. In synthetic data, only Graph-

---

[5]https://archive.ics.uci.edu/ml/datasets/parkinsons+ telemonitoring

PRI and LS are able to further reduce test error. For Graph-PRI, this phenomenon occurs at the beginning of pruning edges, which indicates that our method begins to remove less-informative or spurious connections in an early stage. In Parkinson's data, most of methods obtain almost similar performances to "no edge pruning" (with Graph-PRI performs slightly better as shown in the zoomed plot), which suggests the existence of redundant task relationships. One should note that, the performance of Graph-PRI becomes worse if we remove large amount of edges. One possible reason is that when $|E_s|$ is small, our subgraph tends to have a high graph centrality or star shape, such that one task dominates. Note however that, in MTL, keeping a very sparse relationship is usually not the goal. Because it may lead to weak collaboration between tasks, which violates the motivation of MTL.

## 4.3 fMRI-DERIVED BRAIN NETWORK CLASSIFICATION AND INTERPRETABILITY

Brain networks are complex graphs with anatomic brain regions of interest (ROIs) represented as nodes and functional connectivity (FC) between brain ROIs as links. For resting-state functional magnetic resonance imaging (rs-fMRI), the Pearson's correlation coefficient between blood-oxygen-level-dependent (BOLD) signals associated with each pair of ROIs is the most popular way to construct FC network [Farahani et al., 2019].

In the problem of brain network classification, the identification of predictive subnetworks or edges is perhaps one of the most important tasks, as it offers a mechanistic understanding of neuroscience phenomena [Wang et al., 2021]. Traditionally, this is achieved by treating all the connections (i.e., the Pearson's correlation coefficients) of FC as a long feature vector, and applying feature selection techniques, such as LASSO [Tibshirani, 1996] and two-sample t-test, to determine if one edge connection is significantly different in different groups (e.g., patients with Alzheimer's disease with respect to normal control members).

In this section, we develop a new graph neural networks (GNNs) framework for interpretable brain network classification that can infer brain network categories and identify the most informative edge connections, in a joint end-to-end learning framework. We follow the motivation of [Cui et al., 2021] and aim to learn a global shared edge mask $M$ to highlight decision-specific prominent brain network connections. The final explanation for an input graph $G_i$ is generated by the element-wise product of $A_i$ and $\sigma(M)$, i.e., $A_i \odot \sigma(M)$, in which $A_i$ is the adjacency matrix of $G_i$, $\sigma$ refers to the sigmoid activation function that maps $M$ to $[0, 1]^{N \times N}$. Obviously, $\sigma(M)$ in our GNN also plays a similar role to the edge selection vector $\mathbf{w}$ in Graph-PRI.

**Problem definition.** Given a weighted brain network $G =$ $(V, E, W)$, where $V = \{v_i\}_{i=1}^N$ is the node set of size $N$ defined by the ROIs, $E$ is the edge set, and $W \in \mathbb{R}^{N \times N}$ is the weighted adjacency matrix describing FC strengths between ROIs, the model outputs a prediction label $y$. In brain network analysis, $N$ remains the same across subjects.

**Experimental data.** We evaluate our method on two benchmark real-world brain network datasets. The first one is the eyes open and eyes closed (EOEC) dataset [Zhou et al., 2020], which includes 96 brain networks with the goal to predict either eyes open or eyes closed states. The second one is from the Alzheimer's Disease Neuroimaging Initiative (ADNI) database[6]. We use the brain networks generated by [Kuang et al., 2019], with the task of distinguishing mild cognitive impairment (MCI)[7] group (38 patients) from normal control (NC) subjects (37 in total). Details on brain network construction are elaborated in Appendix C.

**Methodology and objective.** Following [Cui et al., 2021], we provide interpretability by learning an edge mask $M \in \mathbb{R}^{N \times N}$ that is shared across all subjects to highlight the disease-specific prominent ROI connections. Motivated by the functionality of PRI to prune redundant or less informative edges as demonstrated in previous sections, we train $M$ such that the resulting subgraph $G' = G \odot \sigma(M)$ and the original graph $G$ meets the PRI constraint, i.e., Eq. (8). Therefore, the final objective of our interpretable GNN can be formulated as:

$$\mathcal{L}_{\text{CE}} + \lambda \mathbb{E}_{G \sim p(G)} \left\{ S_{\text{vN}}(G') + \beta D_{\text{QJS}}(G'||G) \right\}, \quad (23)$$

in which $\mathcal{L}_{\text{CE}}$ refers to the supervised cross-entropy loss for label prediction, $\lambda$ is the hyperparameter that balances the trade-off between $\mathcal{L}_{\text{CE}}$ and PRI constraint.

**Empirical results.** We summarize the classification accuracy (%) with different methods over 10 independent runs in Table 1, in which Graph-PRI* refers to our objective implemented by approximating von Neumann entropy with Shannon discrete entropy functional on the normalized degree of nodes (see Section 3.4). As can be seen, our method achieves compelling or higher accuracy in both datasets.

To evaluate the interpretability of our method, we visualize the edges been frequently selected for MCI patients and NC group in Fig. 7. We observed that the interactions within sensorimotor cortex (colored blue) for MCI patients are stronger than that of NC group. This result is consistent with the findings in [Ferreri et al., 2016, Niskanen et al., 2011] which observed that the motor cortex excitability is enhanced in AD and MCI from the early stages. We also observed that the interactions within the frontoparietal network (colored yellow) of patients are significantly less than that of NC group, which is in line with previous studies [Neufang et al., 2011, Zanchi et al., 2017] stated that decreased activation in FPN is associated with subtle cognitive deficits.

---

[6]http://adni.loni.usc.edu/

[7]MCI is a transitional stage between AD and NC.

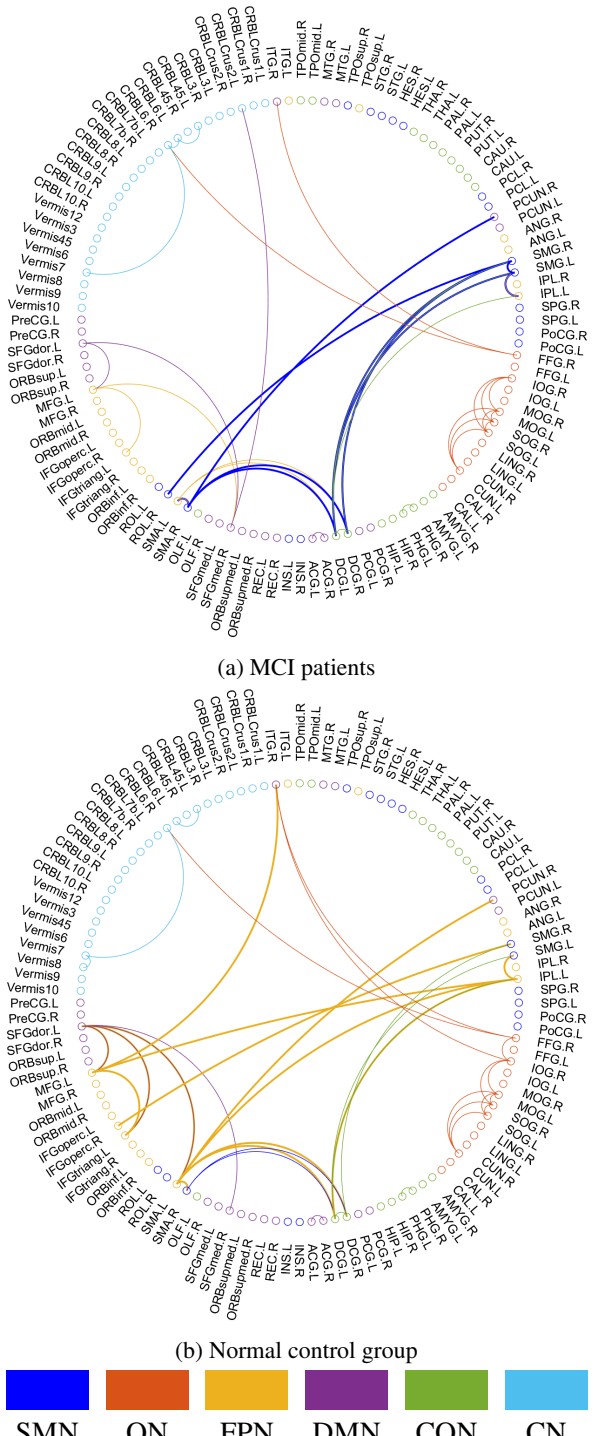

(a) MCI patients

(b) Normal control group

| | SMN | ON | FPN | DMN | CON | CN |

Figure 7: The contributing functional connectivity links for (a) MCI patients; and (b) normal control group. We visualize edges with a probability of more than 50% been selected by our generated edges. The colors of neural systems are described as: sensorimotor network (SMN), occipital network (ON), fronto-parietal network (FPN), default mode network (DMN), cingulo-opercular network (CON), and cerebellum network (CN), respectively.

Table 1: Classification accuracy (%) and standard deviation with different methods over 10 independent runs. The best and second-best performances are in bold and underlined, respectively.

| Method | EOEC | ADNI |
|---|---|---|
| SVM + t-test | $71.79 \pm 7.80$ | $60.61 \pm 10.52$ |
| SVM + LASSO | $72.08 \pm 7.29$ | $54.67 \pm 12.88$ |
| GCN [Kipf and Welling, 2017] | $68.42 \pm 8.59$ | $\mathbf{66.67} \pm 2.48$ |
| GAT [Veličković et al., 2018] | $73.68 \pm 8.60$ | $\mathbf{66.67} \pm 9.43$ |
| **Graph-PRI** | $\mathbf{80.70} \pm 9.60$ | $\mathbf{66.67} \pm 6.67$ |
| **Graph-PRI*** | $\underline{78.95} \pm 4.30$ | $\underline{64.44} \pm 3.14$ |

# 5  CONCLUSIONS

We present a first study on extending the Principle of Relevant Information (PRI) - a less well-known but promising unsupervised information-theoretic principle - to network analysis and graph neural networks (GNNs). Our Graph-PRI preserves spectral similarity well, while also encouraging the resulting subgraph to have higher graph centrality. Moreover, our Graph-PRI is easy to optimize. It can be flexibly integrated with either multi-task learning or GNNs to improve not only the quantitative accuracy but also interpretability.

In the future, we will explore more unknown properties behind Graph-PRI, including a full understanding to the physical meaning of von Neumann entropy on graphs. We will also investigate more downstream applications of Graph-PRI on GNNs such as node representation learning.

### Acknowledgements

The authors would like to thank the anonymous reviewers for constructive comments. The authors would also like to thank Prof. Benjamin Ricaud (at UiT) and Mr. Kaizhong Zheng (at Xi'an Jiaotong University) for helpful discussions. This work was funded in part by the Research Council of Norway (RCN) under grant 309439, and the U.S. ONR under grants N00014-18-1-2306, N00014-21-1-2324, N00014-21-1-2295, the DARPA under grant FA9453-18-1-0039.

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
