# OpenReview forum: "Principle of Relevant Information for Graph Sparsification"
_auai.org/UAI/2022/Conference — UAI 2022 Poster_

### Official Review · Reviewer_DLJW · 2022-03-23

**Q2(1) Originality/Novelty:** 3
**Q2(2) Significance/Impact:** 2
**Q2(3) Correctness/Technical Quality:** 2
**Q2(6) Clarity Of Writing:** 1
**Q6 Overall Score:** 6
**Q8 Confidence In Your Score:** 3

**Q1 Summary And Contributions:**

The paper proposes the principle of relevant information (PRI) for graph sparsification. Specifically, the authors propose to measure information via von Neumann entropy computed from the normalized graph Laplacians, and to optimize an edge selection vector using the proposed PRI. They demonstrate the effectiveness of their approach on graph sparsification tasks, multi-task learning, and network classification tasks.

**Q2 Assessment Of The Paper:**

More detailed information regarding each of these aspects is given below:

**Q2(4) Quality Of Experiments (Optional):**

2: Fair: The experimental evaluation is weak: important baselines are missing, or the results do not adequately support the main claims.

**Q2(5) Reproducibility:**

3: Good: Key resources (e.g., proofs, code, data) are available and key details (e.g., proofs, experimental setup) are sufficiently well-described for competent researchers to confidently reproduce the main results.

**Q3 Main Strengths:**

The paper discusses the PRI for graph sparsification, which to the best of my knowledge is novel. Instantiated using von Neumann entropy and quantum JS divergence (which is based on von Neumann entropy), the PRI is computed from the Laplacians of the original and sparsified graph, respectively. From this perspective, the training objective is internally consistent. The rationale for the choice of the objective is clear and the experimental results are promising, especially the one in Sec. 4.3. I particularly appreciate that the authors make dedicated efforts to justify the PRI for graph sparsification, where they rely on theoretical arguments rather than on experimentation.

**Q4 Main Weakness:**

- The paper requires proofreading by a native speaker. Grammar and spelling is sometimes a bit odd, and this lowers the readability. This is just a superficial comment, though, and not one that influences my rating of the technical content.
- More problematic is the slightly inconsistent use of notation, see 1) below.
- While I appreciate the technical Section 3.2, I question the validity of Assumption 1, which consequently limits the impact of this section. See 2) below.
- It is not clear how the approximation of $S(L)$ by $H(G)$ affects the optimization. Is the gradient still available in closed form, or can the Gumbel-softmax trick still be applied?

**Q5 Detailed Comments To The Authors:**

1) There is a conflict between eqs. (8) and (22). Also, the Laplacians are sometimes denoted as L_G, sometimes with greek letters. In Th. 3, w is used as a lower case letter, while in most of the other instances it is a boldfaced vector. What do $\tilde{1}_M$ and $\tilde{w}_j$ refer to in Th. 3? Is the $\mathbf{w}$ in Section 4.2 the same as the edge selection vector? In Assumption 1, you require strict monotonicity but claim mathematically only monotonicity $\ge$.
2) It may be that Assumption 1 is satisfied in practice (cf. Fig. 1), but I am not sure if this is sufficient to base a theoretical result on it. Also, in the proof of Cor. 1, Lemma 1 makes certain restriction on the increase of the $\lambda_j$, and it is not shown in the proof that these restrictions are met. This somewhat limits the contribution of this section. I do appreciate the experimental evidence in Fig. 1 and 2, respectively.
3) In Sec. 3.2.2, why can for $\beta=0$ the objective be interpreted as $H(G)$? Th. 2 appears to be not sufficient for this claim. Also, the transition of how the results from $H(G)$ carry over to $S(L)$ is not clear in this paragraph.
4) In Sec. 4.1, it is wrongly claimed that the aim of the method is to "preserve centrality". This is not the case. Rather, the aim should be to preserve spectral similarity, and to trade this for increased centrality.
5) In Sec. 4.2, it is not clear why the sparsified graph should improve generalization performance. A few lines pointing at negative transfer or similar phenomena could help to justifiy this assumption.
6) The figures need to be improved. I would expect figures to display functions (every x value has a single y value), and it appears that this is not the case for some of the figures. Please rearrange the order of datapoints accordingly, or explain why the figures need to look like this.


EDIT: Score updated based on authors' response.

**Q7 Justification For Your Score:**

I like the general idea of the paper, but I think that the paper is not yet ready for publication. The experiments are nice and I appreciate the theoretical discussions, but the presentation is currently subpar. Also, some of the theoretical claims require additional justification.

**Q9 Complying With Reviewing Instructions:**

1: Yes.

---

### Official Review · Reviewer_pk4K · 2022-03-29

**Q2(1) Originality/Novelty:** 3
**Q2(2) Significance/Impact:** 2
**Q2(3) Correctness/Technical Quality:** 2
**Q2(6) Clarity Of Writing:** 3
**Q6 Overall Score:** 6
**Q8 Confidence In Your Score:** 3

**Q1 Summary And Contributions:**

The paper proposes a graph sparsification method based on a generalization of the principle of relevant information to graph Laplacians. It motivates why this new method is useful with both analytical and empirical results.


**Q2 Assessment Of The Paper:**

More detailed information regarding each of these aspects is given below:

**Q2(4) Quality Of Experiments (Optional):**

3: Good: The experimental evaluation is adequate, and the results convincingly support the main claims.

**Q2(5) Reproducibility:**

3: Good: Key resources (e.g., proofs, code, data) are available and key details (e.g., proofs, experimental setup) are sufficiently well-described for competent researchers to confidently reproduce the main results.

**Q3 Main Strengths:**

Graph sparsification is a practically relevant problem. The method for graph sparsification proposed by the paper, graph-PRI, is elegant and flexible, as is illustrated by the simulations and real data examples given in the paper. The paper is also well structured and overall easy to follow.



**Q4 Main Weakness:**

The paper has three major weaknesses. The first is that Graph-PRI is develped based on the van-Neumann-entropy and its properties but
Section 3.4 indicates that in practice it is approximated with Shanon entropy throughout (see also Section 3.2.2 and the case with beta=0 where vNE is also seemingly replaced with the SE). This is motivated by Theorem 2 which is in my opinion too little motiviation on its own. The second issue is how the results from Section 3.2.1 are mathematically formulated. I suppose that Assumption 1 will generally not hold for all edges I may add to G and in that sense it most certainly not a mild assumption. Based on Theorem 1 it is, however, in some sense likely for Assumption 1 to hold for most edges. The way Section 3.2.1 is formulated does not always reflect this subtlety (see for example the certainty with which the paragraph following the Corollary is formulated). Third, there are issues with the correctness of the English throughout the paper.

**Q5 Detailed Comments To The Authors:**

- Bold to say that it is "always prohibitive to exactly visualize analyze a graph" in the first paragraph of the introduction.
- Missing "are" in the second paragraph on page 2.
- The last two paragraph of Section 2.1 both have grammar issues.
- Who are "authors" at the top of page 3 and is there a comma missing between "community detection" and "Local similarity"?
- In section 2.2 I don't understand what the difference is between the broad existing literature on graph sparsification and learning-based algorithms for graph sparsification which apparently has not been considered much in the literature.
- What does the first sentence on the right column of page 3 mean.
- The paragraph before Theorem 2 is hard to read due to grammar issues. Also what is the weight matrix in Theorem 2 the weight matrix of? The graph I
suppose but this is never stated.
- Bottom of page 4: what is "the spectral property"?
- Section 3.3: first paragraph has writing issues and also should it be equation 11 in Theorem 3?
- Section 3.4: "comprising" should probably be compromising
- Section 4.1: How do Graph-PRI and LD compare in terms of their computational burden? It would be good to add a few sentences on this.

**Q7 Justification For Your Score:**

The paper considers an interesting problem and proposes an elegant and flexible solution to it. There are some issues with the presentation of certain aspects of the results and the writing, however, which lowers my score by a bit.

**Q9 Complying With Reviewing Instructions:**

1: Yes.

---

### Official Review · Reviewer_w83v · 2022-04-12

**Q2(1) Originality/Novelty:** 3
**Q2(2) Significance/Impact:** 2
**Q2(3) Correctness/Technical Quality:** 3
**Q2(6) Clarity Of Writing:** 3
**Q6 Overall Score:** 6
**Q8 Confidence In Your Score:** 3

**Q1 Summary And Contributions:**

The paper addresses the problem of graph sparsification problem that aims to reduce the number of edges of a graph while maintaining its structural properties. To do so, it adapts the Principle of Relevant Information (PRI) to graphs. Experiments illustrate the behaviour of the proposed approach.

Overall, the idea is interesting, its application to graphs is novel, and the paper is well written.

**Q10 Ethical Concerns (Optional):**

none.

**Q2 Assessment Of The Paper:**

More detailed information regarding each of these aspects is given below:

**Q2(4) Quality Of Experiments (Optional):**

3: Good: The experimental evaluation is adequate, and the results convincingly support the main claims.

**Q2(5) Reproducibility:**

3: Good: Key resources (e.g., proofs, code, data) are available and key details (e.g., proofs, experimental setup) are sufficiently well-described for competent researchers to confidently reproduce the main results.

**Q3 Main Strengths:**

The idea is interesting, its application to graphs is novel, and the paper is well written. Experiments illustrate the behaviour of the proposed approach.

**Q4 Main Weakness:**

I'm wondering why in the problem formulation, sect. 2.1, in G_s, one does not consider also the case to reduce the node set to V_s \subseteq  V.


**Q5 Detailed Comments To The Authors:**

Minor:

in def. of G_s, E_s \in E -> E_s \subseteq E


**Q7 Justification For Your Score:**

Overall, the idea is interesting, its application to graphs is novel, and the paper is well written. Experiments illustrate the behaviour of the proposed approach.

**Q9 Complying With Reviewing Instructions:**

1: Yes.

---

### Decision · Program_Chairs · 2022-05-15

**Decision:**

Accept (Poster)

**Comment:**

Meta Review: This paper proposes a new method to sparsify an (undirected) graph by optimizing an objective function based on a principle of relevant information. The rationale for and properties of the objective objection are fairly well explained, and the empirical performance of the method is illustrated in experiments, some of which are quite interesting and practically relevant.

The three reviewers converged on a verdict of weak accept. One common concern is with Assumption 1 and the subsequent Corollary 1. In my view (which seems to be also indicated by one of the reviewers), it is probably better to reformulate Corollary 1 by incorporating assumption 1 as a condition on the added edge, something like: if the added edge satisfies assumption 1, then the QJS divergence from the original graph will decrease. Then one can use Theorem 1 and empirical evidence to suggest that this condition is probably satisfied by most edges. Two more issues about Corollary 1 that I think the authors should clarify in their revisions. First, the proof seems to establish that the result holds *approximately* rather than strictly. Is this true? Second, Corollary allows any edge to be added, including edges that are not in the original graph. Is that intended? It seems to me that the added edge should be confined to those in the original graph, so that the resulting graph remains a subgraph of the original graph and Corollary 1 can then be applied repeatedly to show that a less sparse subgraph tends to have a lower QJS.